# Characterization of patients receiving surgical versus non-surgical treatment for infective endocarditis in West Virginia

**Ruchi Bhandari[1] *, Noor Abdulhay[1], Talia Alexander[1,2], Jessica Rubenstein[1], Andrew Meyer[1], Frank H. Annie[3], Umar Kaleem[4], R. Constance Wiener[5], Cara Sedney[6], Ellen Thompson[4], Affan Irfan[7]**

**1** School of Public Health, West Virginia University, Morgantown, WV, United States of America, **2** National Institute for Occupational Safety and Health, Centers for Disease Control and Prevention, Atlanta, Georgia, United States of America, **3** Health Education and Research Institute, Charleston Area Medical Center, Charleston, West Virginia, United States of America, **4** Joan C. Edwards School of Medicine, Marshall University, Huntington, WV, United States of America, **5** School of Dentistry, West Virginia University, Morgantown, WV, United States of America, **6** School of Medicine, West Virginia University, Morgantown, WV, United States of America, **7** Mayo Clinic Health System, Rochester, MN, United States of America

* rbhandari@hsc.wvu.edu

**Data Availability Statement:** Data underlying this study are from an approved repository that houses clinical data from the four healthcare systems in

## Abstract

### Background

Infective endocarditis (IE) has increased in rural states such as West Virginia (WV) with high injection drug use. IE is medically managed with antimicrobial treatment alone or combined with surgical treatment. This study aimed to characterize the predictors associated with surgical treatment and rates of inpatient mortality and readmission among IE patients in WV's rural centers.

### Methods

This retrospective review of electronic health records includes all adults hospitalized for IE at major rural tertiary cardiovascular centers in WV during 2014–2018. Descriptive statistics were presented on demographics, history of injection drug use, clinical characteristics, and hospital utilization by surgery status, and multivariable logistic regression examined the association of surgery with key predictor variables, generating odds ratios (OR).

### Results

Of the 780 patients with IE, 38% had surgery, with a 26-fold increase in patients undergoing surgery between 2014–2018. Comparing surgery and non-surgery patients revealed significant differences. Surgery patients were significantly younger (median age 35.6 vs. 40.5 years; p<0.001); had higher rates of drug use history (80% vs. 65%; p<0.001), psychiatric disorders (57% vs. 31%; p<0.001), and readmissions (18% vs.12%; p = 0.015). Surgery patients had lower rates of discharge against medical advice (11% vs.17%; p = 0.028) and in-hospital mortality (5% vs.12%; p<0.001). In the multivariable logistic regression, surgery was associated with injection drug use (OR: 1.9; 95% CI:1.09–3. 3), indications for surgery

West Virginia. These data contain full Protected Health Information (PHI) and thus, legally cannot be shared publicly. Data are available upon request from Wes Kimble (wkimble1@hsc.wvu.edu), Director of Research Data Analytics, West Virginia Clinical and Translational Science Institute, for researchers who meet the criteria for access to confidential data.

**Funding:** The source of funding for our study was: United States National Institute of General Medical Sciences [Grant number: 2U54GM104942-07]. The funders had no role in study design, data collection and analysis, decision to publish, or preparation of the manuscript. The authors received no salary from the funding organization for this work.

**Competing interests:** The authors have declared that no competing interests exist.

(OR: 1.68; 95% CI:1.48–1.91), left-sided IE (OR: 2.14; 95%CI:1.43–3.19) and later years (OR:3.75; 95%CI:2.5–5.72).

## Conclusion

This study characterizes the predictors associated with surgical treatment and rates of inpatient mortality and readmission among IE patients across rural WV. The decision to perform cardiac surgery on IE patients is complex. Results with increased injection drug use-associated IE emphasize the importance of comprehensive care by a multidisciplinary team for optimal management of patients with IE.

## Introduction

Infective endocarditis (IE) is a severe infection of the endocardium (inner lining of the heart and/or valves) that affects 15 per 100,000 people in the United States [1], and the incidence is increasing steeply. In a recent multicenter retrospective study of electronic health records (EHR), there was an increase of 458% in patients hospitalized for IE in West Virginia (WV) during 2014–2018 [2].

IE is medically managed either with only antimicrobial treatment or a combination of antimicrobial treatment with surgical treatment [3]. The American Heart Association, the American College of Cardiology, and the European Society of Cardiology have developed evidence-based guidelines where surgical treatment is recommended in addition to antimicrobial treatment, based upon several factors, including surgical indications, heart failure and shock, microorganisms and persistent bacteremia, embolic risk reduction, right-sided IE, operative risk assessment, and risk of IE relapse [4–6].

The benefits of surgical treatment of IE under such recommendations have been demonstrated in several research studies. Surgical treatment of IE has been found to be associated with prevention of embolic sequelae and reduction of systemic embolism [6], and lower mortality in specific populations [7]. Surgery has also shown to prevent destruction of the valves and facilitate rapid recovery in patients with IE [8]. In patients who inject drugs, valve repair is preferred over replacement to preserve valve function and avoid foreign material implant [3]. However, there can be many pre- or post-operative complications in patients with injection drug use-associated IE (DU-IE), including heart failure, reinfection, neurological complications [9], stroke [10], hemodialysis, as well as prolonged respiratory failure requiring tracheostomy [8].

In the past decade, there has been a significant shift with injection drug use as the major risk factor for IE. Drug overdoses and deaths have increased sharply in the United States between 2014 and 2022, with West Virginia having the highest age-adjusted drug overdose mortality (81.4 per 100,000) [11]. Cardiac surgeries for the treatment of DU-IE have increased in tandem with the opioid epidemic in West Virginia [12]. Concomitantly, IE-related mortality has increased in new population groups, particularly in rural patients [13]. The approaches to manage and treat IE also evolve and refine with the changing risk factors [3, 14].

Given the changing epidemiology of the patient population, the purpose of this research study was to characterize predictors among patients hospitalized with IE stratified by those who received only antimicrobial treatment versus those who received antimicrobial and surgical treatment in the four major rural centers in West Virginia. We further describe the surgical characteristics and outcomes in the subpopulation of patients who received surgical treatment.

This report is of particular significance because the characteristics and outcomes with respect to treatment stratification have not previously been described for patients at rural centers.

## Materials and methods

This study is a retrospective chart review of EHR of all adults between the ages of 18 to 90 years who had their first IE hospital admission between January 1, 2014, and December 31, 2018, in four university-affiliated referral hospitals in WV. The hospitals in the study are the only tertiary centers in the state with the capability to perform heart surgery, and patients are referred to these centers from across the entire state [2]. Patients were identified initially using the International Classification of Diseases, Tenth Revision, Clinical Modification (ICD-10-CM) codes for IE [2], followed by a manual chart review for all admissions to ensure accurate diagnosis confirmation. Data were captured in a secure, HIPAA-compliant, web-based system using the Research Electronic Data Capture (REDCap) [15]. During chart review, we extracted information of individual patients from history and physical examination notes, provider notes, operative notes, consultation notes, hospital narratives, laboratory tests and imaging results, and discharge summaries. The study was approved by the Institutional Review Board at West Virginia University (IRB protocol number: 1811373348). No written, signed consent was required for this retrospective study of EHR as a Health Insurance Portability and Accountability Act (HIPAA) waiver of authorization was obtained. When data were entered from EHR into REDCap, identifiable information was available. A deidentified database was created for the purpose of analyses.

The outcome variable is patients with IE undergoing surgery (along with antimicrobial treatment) vs. no surgery (antimicrobial treatment alone) at patient's index admission for IE. Descriptive characteristics are presented on (a) demographics: sex (male/female); age (18–44, 45–64, ≥65 years); (b) substance use: smoking status (current/former/non-smoker); alcohol use (current/former/no use); injection drug use (yes/no); (c) clinical characteristics: comorbidities; number of comorbidities; psychiatric disorders; affected valve (Tricuspid/Mitral/Aortic/Pulmonic); causative organisms (methicillin-resistant *Staphylococcus aureus* [MRSA]/methicillin-susceptible *Staphylococcus aureus* [MSSA]/other); indications for surgery: valvular regurgitation (trace/mild/moderate/severe), vegetation size in each valve (diffuse thickening/small/medium/large), and embolism type; and (d) hospital utilization: consultations; length of hospital stay; length of Intensive Care Unit stay; readmission; and discharge status (alive/against medical advice (AMA)/death). The discharge categories are mutually exclusive. In addition, the following data were collected for the patients who had surgery: the valve involved (aortic, mitral, tricuspid, pulmonic), valve intervention (repair versus replacement), surgical approach (sternotomy or minimally invasive right thoracotomy), myocardial protection strategy (cardioplegia versus beating heart), and concomitant procedures.

Data were obtained for the first admission of each patient during the study period. In addition, information was also recorded on patient's readmission during the study period. Categorical variables are presented as counts and percentages. Surgery and non-surgery groups were compared using Chi-square test or Fisher's exact test when expected cell count was <5. Continuous variables are presented as median and interquartile range. Statistical analyses were conducted using R version 4.0.2 (R Foundation for Statistical Computing) and SPSS version 28.0. Statistical significance was accepted at $p < 0.05$. Adjustments were made using Bonferroni correction wherever multiple tests were conducted. Highlighted p-values reflect statistical significance in the tables after Bonferroni correction.

Multivariable logistic regression analysis was conducted to examine the association between the key dependent variable, surgery (yes/no), and key predictor and potentially confounding

variables: age (continuous), injection drug use (dichotomous), valve (right-sided/left-sided/bilateral), number of indications for surgery (ordinal), and years (2014-2016/2017/2018). Other variables, such as sex and race, were not included in the analysis because they were not significant in the bivariate analysis.

## Results

Of the 780 patients with IE who were admitted between January 1, 2014, and December 31, 2018, 37.9% had surgery. During this period, there was a 2.5-fold increase in patients who were treated non-surgically and a 26-fold increase in patients who underwent surgery as part of their management (p < 0.001) (Fig 1). The sample characteristics of the patients stratified by surgery vs. no surgery are presented in Table 1. Patients with surgery were much younger, with 71.5% in the 18–44 age group (median age of 35.6 years) compared with 58.9% of patients without surgery (median age of 40.5 years) (p < 0.001). Male and female patients hospitalized for IE during this period did not differ by surgery status. A significantly higher proportion of patients with surgery reported being current smokers (73.9% vs. 57.9%; p < 0.001), having injected drugs prior to hospital admission (80.3% vs. 65.1%; p < 0.001), and being on medications for opioid use disorder (MOUD) prior to hospital admission (33.9% vs. 16.5%; p < 0.001) compared to patients without surgery. A significantly higher proportion of patients with surgery also used opioids, amphetamines, cannabinoids, cocaine metabolites, and benzodiazepines (all p < 0.001).

Clinical characteristics of patients with IE stratified by surgery status are presented in Table 2. A lower proportion of patients with surgery had comorbidities, including Type 2 diabetes, coronary artery disease, hyperlipidemia, acute kidney injury, and peripheral vascular disease compared to patients without surgery. However, a higher proportion of patients with surgery were diagnosed with psychiatric disorders (57.3% vs. 31.0%) including, substance use disorder (45.1% vs. 19.0), depression (28.8% vs 15.9%), and anxiety (27.5% vs. 12.4%) (all p < 0.001), compared to patients without surgery.

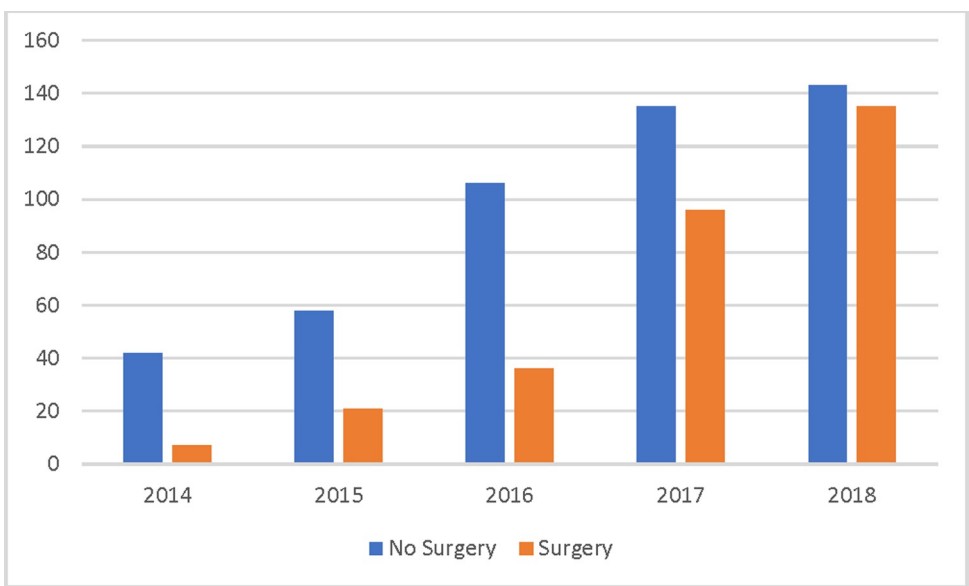

**Fig 1. Infective endocarditis patients with and without surgery by year.**

**Table 1. Baseline characteristics of patients with infective endocarditis by surgery status.**

| | Surgery | | No Surgery | | p-value |
|---|---|---|---|---|---|
| Total | N = 295 | 37.9% | N = 484 | 62.1 | |
| | N | % | N | % | |
| **Years** | | | | | **< 0.001** |
| 2014 | 5 | 1.7 | 41 | 8.5 | |
| 2015 | 21 | 7.1 | 58 | 12 | |
| 2016 | 36 | 12.2 | 106 | 21.9 | |
| 2017 | 96 | 32.5 | 135 | 27.9 | |
| 2018 | 135 | 45.8 | 143 | 29.5 | |
| Missing | 2 | 0.7 | 1 | 0.2 | |
| **Sex** | | | | | 0.524 |
| Male | 152 | 51.5 | 238 | 49.2 | |
| Female | 143 | 48.5 | 246 | 50.8 | |
| **Age** | | | | | **< 0.001** |
| 18–44 | 211 | 71.5 | 285 | 58.9 | |
| 45–64 | 58 | 19.7 | 114 | 23.6 | |
| 65+ | 25 | 8.5 | 85 | 17.6 | |
| Missing | 1 | 0.3 | 0 | 0 | |
| **Age (Median and IQR)** | 35.6 | 17.7 | 40.5 | 27.2 | **< 0.001** |
| **Smoking status** | | | | | **< 0.001** |
| Current smoker | 218 | 73.9 | 280 | 57.9 | |
| Former smoker | 47 | 15.9 | 65 | 13.4 | |
| Non-smoker | 22 | 7.5 | 76 | 15.7 | |
| Missing | 8 | 2.7 | 63 | 13.0 | |
| **Alcohol use** | | | | | **0.009** |
| Current alcohol use | 74 | 25.1 | 80 | 16.5 | |
| Former alcohol use | 38 | 12.9 | 30 | 6.2 | |
| No alcohol use | 157 | 53.2 | 251 | 51.9 | |
| Missing | 26 | 8.8 | 123 | 25.4 | |
| **Injection drug use prior to hospital admission** | | | | | **< 0.001** |
| Yes | 237 | 80.3 | 315 | 65.1 | |
| No | 57 | 19.3 | 157 | 32.4 | |
| Missing | 1 | 0.3 | 12 | 2.5 | |
| **Type of drug** | | | | | |
| Opiates | 203 | 68.8 | 274 | 56.6 | **< 0.001** |
| Amphetamines | 98 | 33.2 | 87 | 18.0 | **< 0.001** |
| Cannabinoids (marijuana) | 97 | 32.9 | 81 | 16.7 | **< 0.001** |
| Buprenorphine | 80 | 27.1 | 61 | 12.6 | **< 0.001** |
| Cocaine metabolites | 77 | 26.1 | 63 | 13.0 | **< 0.001** |
| Benzodiazepines | 53 | 18.0 | 40 | 8.3 | **< 0.001** |
| Methadone | 21 | 7.1 | 12 | 2.5 | **0.002** |
| **MOUD\* prior to hospital admission** | | | | | |
| Yes | 100 | 33.9 | 80 | 16.5 | **< 0.001** |
| No | 139 | 47.1 | 313 | 64.7 | |
| Missing | 56 | 19.0 | 91 | 18.8 | |

\*MOUD: Medications for opioid use disorder

Note: Highlighted p-values reflect statistical significance after Bonferroni correction.

**Table 2. Clinical characteristics of patients with infective endocarditis by surgery status.**

| | Surgery | | No Surgery | | p-value |
|---|---|---|---|---|---|
| Total | N = 295 | 37.9% | N = 484 | 62.1 | |
| | **N** | **%** | **N** | **%** | |
| **Comorbidities** | | | | | |
| Psychiatric disorders | 169 | 57.3 | 150 | 31.0 | **< 0.001** |
| Hypertension | 82 | 27.8 | 148 | 30.6 | 0.409 |
| Type 2 Diabetes | 34 | 11.5 | 89 | 18.4 | 0.011 |
| Coronary artery disease | 22 | 7.5 | 73 | 15.1 | **0.002** |
| Chronic lung disease | 32 | 10.8 | 56 | 11.6 | 0.757 |
| Hyperlipidemia | 21 | 7.1 | 64 | 13.2 | 0.008 |
| Acute kidney injury | 20 | 6.8 | 57 | 11.8 | 0.023 |
| Chronic kidney disease | 22 | 7.5 | 48 | 9.9 | 0.244 |
| Stroke | 23 | 7.8 | 38 | 7.9 | 0.978 |
| Peripheral vascular disease | 13 | 4.4 | 44 | 9.1 | 0.015 |
| Metastatic infections | 11 | 3.7 | 28 | 5.8 | 0.202 |
| Cancer | 9 | 3.1 | 28 | 5.8 | 0.082 |
| **Psychiatric disorders** | | | | | |
| Substance use disorder | 133 | 45.1 | 92 | 19.0 | **< 0.001** |
| Depression | 85 | 28.8 | 77 | 15.9 | **< 0.001** |
| Anxiety | 81 | 27.5 | 60 | 12.4 | **< 0.001** |
| Bipolar disorder | 15 | 5.1 | 25 | 5.2 | 0.961 |
| Post-traumatic stress disorder | 18 | 6.1 | 17 | 3.5 | 0.091 |
| **Type of IE** | | | | | |
| Tricuspid | 143 | 48.5 | 209 | 43.2 | 0.150 |
| Mitral | 102 | 34.6 | 132 | 27.3 | 0.031 |
| Aortic | 91 | 30.8 | 108 | 22.3 | **0.008** |
| Pulmonic | 4 | 1.4 | 19 | 3.9 | 0.040 |
| **Causative organisms** | | | | | |
| MRSA - *Staphylococcus aureus*, methicillin resistant | 106 | 35.9 | 227 | 46.9 | **0.003** |
| MSSA - *Staphylococcus aureus*, methicillin sensitive | 88 | 29.8 | 115 | 23.8 | 0.061 |
| *Streptococci* species | 33 | 11.2 | 36 | 7.4 | 0.074 |
| Other | 20 | 6.8 | 44 | 9.1 | 0.255 |
| Enterococcus species | 27 | 9.2 | 33 | 6.8 | 0.236 |
| Candida species | 25 | 8.5 | 23 | 4.8 | 0.036 |
| Serratia species | 23 | 7.8 | 19 | 3.9 | 0.020 |
| Culture negative | 13 | 4.4 | 18 | 3.7 | 0.743 |
| Viridians Streptococci | 11 | 3.7 | 15 | 3.1 | 0.635 |
| Klebsiella species | 9 | 3.1 | 7 | 1.4 | 0.126 |
| **Indications for surgery** | | | | | |
| Vegetation size | 282 | 95.6 | 414 | 85.5 | **< 0.001** |
| Valvular regurgitation | 252 | 85.4 | 257 | 53.1 | **< 0.001** |
| Embolism | 163 | 55.3 | 159 | 32.9 | **< 0.001** |
| Persistent sepsis/persistent positive blood cultures | 108 | 36.6 | 101 | 20.9 | **< 0.001** |
| Septic shock | 71 | 24.1 | 75 | 15.5 | **0.003** |
| Stroke | 36 | 12.2 | 51 | 10.5 | 0.474 |
| Acute congestive heart failure | 27 | 9.2 | 14 | 2.9 | **< 0.001** |
| Root abscess | 26 | 8.8 | 6 | 1.2 | **< 0.001** |

(*Continued*)

**Table 2.** (Continued)

| | Surgery | | No Surgery | | p-value |
|---|---|---|---|---|---|
| Cardiogenic shock | 13 | 4.4 | 13 | 2.7 | 0.195 |

Note: Highlighted p-values reflect statistical significance after Bonferroni correction.

A higher percentage of patients who underwent surgery were diagnosed with mitral valve IE (34.6% vs. 27.3%; p = 0.031) and aortic valve IE (30.8% vs. 22.3%; p = 0.008); and fewer patients with surgery had MRSA-IE (35.9% vs. 46.9%; p = 0.003), compared to patients without surgery. Additionally, more patients with surgery had the following indications for surgery: vegetation size (large ≥10mm), valvular regurgitation (mild or moderate), embolism (specifically, pulmonary embolism), persistent sepsis/persistent positive blood cultures, acute congestive heart failure, and root abscess (all p < 0.001). However, stroke and cardiogenic shock were not significantly different between the two groups.

Hospital utilization also varied between patients with and without surgical treatment, presented in Table 3. A higher proportion of patients with surgery utilized the following consultation services: cardiac surgery, general surgery, interventional radiology, dentistry, pain management, social work, physical/occupational therapy, and psychiatry. Of the patients who underwent surgery, 58.7% stayed in the hospital for 30 or more days, compared to 21% of the medically (not surgically) treated patients. Almost half (48.9%) of the patients with surgery were in the intensive care unit for six or more days compared with only 15% of the patients without surgery. Furthermore, a significantly higher proportion of patients with surgery were discharged alive (83.7% vs. 70.7%; p = 0.001); although readmissions were also higher in patients with surgery (18.3% vs. 12.0%; p = 0.015). Significantly fewer patients with surgery were discharged against medical advice (11.2 vs. 16.9%; p = 0.028), and experienced less in-hospital mortality (5.1% vs. 12.4%; p < 0.001), compared to patients without surgery.

The surgical characteristics of patients with IE can be observed in Table 4. Among the patients who underwent surgery, the most common EKG finding was sinus rhythm, which occurred among almost 40% of the patients. Valve replacement/repair was the predominant type of surgery, with almost half (48.6%) of the patients having a tricuspid valve replacement/repair, a third (33.9%) undergoing mitral valve replacement/repair, and about 27.8% undergoing aortic valve replacement/repair. Majority of surgeries (80%) used the median sternotomy approach with arrested heart. Almost a quarter had additional procedures such as drainage of pleural effusion and dental extractions.

In the multivariable logistic regression results presented in Table 5, surgery was statistically significantly associated with injection drug use (OR: 1.895; 95% CI: 1.089–3.299), left-sided valve with reference to right-sided valve (2.136; 95% CI: 1.429–3.193), bilateral valve with reference to right-sided valve (2.473; 95% CI: 1.240–4.932), number of indications for surgery (OR: 1.680; 95% CI: 1.482–1.905), year 2017 with reference to 2014–2016 (OR: 3.147; 95% CI: 2.030–4.877) and year 2018 with reference to 2014–2016 (OR: 3.746; 95% CI: 2.455–5.716).

## Discussion

Our results demonstrate that, compared with the number of IE patients who were only medically managed, the number of patients who were medically managed and underwent surgery increased 26-fold during the study period across four major rural centers in WV. Surgical intervention was more likely among patients who were significantly younger, injected drugs,

**Table 3. Hospital utilizations of patients with infective endocarditis by surgery status.**

| | Surgery | | No Surgery | | p-value |
|---|---|---|---|---|---|
| Total | N = 295 | 37.9% | N = 484 | 62.1 | |
| | **N** | **%** | **N** | **%** | |
| **Consultations** | | | | | |
| Infectious Disease | 291 | 98.6 | 459 | 94.8 | **0.006** |
| Cardiac Surgery | 289 | 98.0 | 359 | 74.2 | **< 0.001** |
| Cardiology | 204 | 69.2 | 344 | 71.1 | 0.569 |
| Social work | 197 | 66.8 | 190 | 39.3 | **< 0.001** |
| Physical/Occupational therapy | 149 | 50.5 | 171 | 35.3 | **< 0.001** |
| Psychiatry | 175 | 59.3 | 108 | 22.3 | **< 0.001** |
| Nephrology | 106 | 35.9 | 162 | 33.5 | 0.483 |
| Dentistry | 141 | 47.8 | 48 | 9.9 | **< 0.001** |
| Spiritual counseling | 86 | 29.2 | 90 | 18.6 | **< 0.001** |
| General Surgery | 74 | 25.1 | 61 | 12.6 | **< 0.001** |
| Neurology | 51 | 17.3 | 79 | 16.3 | 0.726 |
| Vascular | 49 | 16.6 | 65 | 13.4 | 0.223 |
| Orthopedic Surgery | 47 | 15.9 | 57 | 11.8 | 0.098 |
| Pain management | 56 | 19.0 | 19 | 3.9 | **< 0.001** |
| Neurosurgery | 30 | 10.2 | 45 | 9.3 | 0.689 |
| Interventional Radiology | 49 | 16.6 | 26 | 5.4 | **< 0.001** |
| Ophthalmology | 32 | 10.8 | 39 | 8.1 | 0.190 |
| Individual therapy | 14 | 4.7 | 4 | 0.8 | **< 0.001** |
| Music therapy | 5 | 1.7 | 2 | 0.4 | 0.111 |
| **ICU length of stay** | | | | | **< 0.001** |
| 0 hours | 6 | 2.0 | 332 | 68.6 | |
| >0 hours to 2 days | 73 | 24.7 | 40 | 8.3 | |
| 3 days | 30 | 10.2 | 17 | 3.5 | |
| 4–5 days | 42 | 14.2 | 21 | 4.3 | |
| 6–7 days | 35 | 11.9 | 15 | 3.1 | |
| 8–10 days | 31 | 10.5 | 15 | 3.1 | |
| 11 or more days | 77 | 26.1 | 43 | 8.9 | |
| Missing | 1 | 0.3 | 1 | 0.2 | |
| **Discharge status** | | | | | **< 0.001** |
| Discharge alive | 247 | 83.7 | 342 | 70.7 | |
| Discharge AMA* | 33 | 11.2 | 82 | 16.9 | |
| Death | 15 | 5.1 | 60 | 12.4 | |
| **Readmission** | | | | | **0.015** |
| Yes | 54 | 18.3 | 58 | 12.0 | |
| No | 241 | 81.7 | 426 | 88.0 | |
| **Length of Stay** | | | | | **< 0.001** |
| < = 4 days | 5 | 1.7 | 70 | 14.5 | |
| 5–9 days | 11 | 3.7 | 102 | 21.1 | |
| 10–19 days | 11 | 3.7 | 142 | 29.3 | |
| 20–29 days | 57 | 19.3 | 64 | 13.2 | |
| 30–39 days | 33 | 11.2 | 29 | 6.0 | |
| 40–49 days | 69 | 23.4 | 51 | 10.5 | |
| 50 or more days | 71 | 24.1 | 23 | 4.8 | |

(*Continued*)

**Table 3.** (Continued)

| | Surgery | | No Surgery | | p-value |
|---|---|---|---|---|---|
| Missing | 1 | 0.3 | 3 | 0.6 | |

*AMA: Against medical advice

Note: Highlighted p-values reflect statistical significance after Bonferroni correction.

were diagnosed with psychiatric disorders, had mitral valve IE or aortic valve IE, had more indications for surgery, and were in the intensive care unit for an extended period of time.

Findings from our study show a predominantly young population with IE, with median age of 35 years. The difference in age can be attributed to the increasing injection drug use in

**Table 4. Surgical characteristics of patients with infective endocarditis.**

| | N = 295 | % |
|---|---|---|
| **EKG Results** | | |
| Sinus rhythm | 117 | 39.7 |
| Sinus tachycardia | 12 | 4.1 |
| Heart block | 12 | 4.1 |
| Permanent pacemaker required | 12 | 4.1 |
| **Aortic valve replacement/repair** | | |
| Aortic valve replacement (tissue) | 46 | 16.0 |
| Aortic valve replacement (mechanical) | 23 | 8.0 |
| Aortic valve repair | 11 | 3.8 |
| **Mitral valve replacement/repair** | | |
| Mitral valve repair | 42 | 14.5 |
| Mitral valve replacement (tissue) | 42 | 14.5 |
| Mitral valve replacement (mechanical) | 14 | 4.8 |
| **Tricuspid valve replacement/repair** | | |
| Tricuspid valve repair | 73 | 25.3 |
| Tricuspid valve replacement (tissue) | 61 | 21.2 |
| Tricuspid valve replacement (mechanical) | 6 | 2.1 |
| **Surgery heart status** | | |
| Cardioplegia | 236 | 80.0 |
| Beating heart (off pump) | 23 | 7.8 |
| Unknown | 36 | 12.2 |
| **Surgical approach** | | |
| Median sternotomy | 236 | 80.0 |
| Right thoracotomy | 21 | 7.1 |
| Missing | 38 | 12.9 |
| **Concomitant procedures** | | |
| Pleural effusion | 74 | 25.1 |
| Dental extractions (pre- or post-op) | 66 | 22.4 |
| Soft tissue debridement | 19 | 6.5 |
| Patent foramen ovale closure | 16 | 5.4 |
| Tracheostomy | 16 | 5.4 |
| Return to operating room for bleeding | 15 | 5.1 |
| Postop pericardial drain/window | 12 | 4.1 |
| Joint aspiration/debridement | 14 | 4.7 |
| Video-assisted thoracoscopic surgery/Thoracotomy | 10 | 3.4 |

**Table 5. Multivariable logistic regression: Surgery vs. No surgery.**

| Variable | Unadjusted OR | 95% CI | Adjusted OR | 95% CI |
|---|---|---|---|---|
| Age | 0.981 | (0.971,0.990) | 0.979 | (0.964, 0.995) |
| Injection drug use (Y/N) | 2.072 | (1.465, 2.931) | 1.895 | (1.089,3.299) |
| Type of IE - right sided valve | Ref. | Ref. | Ref. | Ref. |
| Type of IE - left sided valve | 2.474 | (1.334,4.587) | 2.136 | (1.429,3.193) |
| Type of IE - bilateral valve | 1.256 | (0.922,1.715) | 2.473 | (1.240,4.932) |
| Indications of surgery | 1.702 | (1.520,1.906) | 1.680 | (1.482,1.905) |
| Year - 2014–2016 | Ref. | Ref. | Ref. | Ref. |
| Year - 2017 | 2.351 | (1.598,3.460) | 3.147 | (2.030,4.877) |
| Year - 2018 | 3.121 | (2.159,4.514) | 3.746 | (2.455,5.716) |

younger age groups in the United States. Patients who inject drugs are more likely to be younger, with fewer comorbidities or predisposing heart conditions [16]. These results are consistent with previous studies that show that the proportion of IE hospitalizations related to injection drug use has continued to increase over the years in the United States [17], and in some specific regions, such as WV [2].

Our results show a greater proportion of patients with surgery were diagnosed with psychiatric disorders, such as substance use disorder. This is not an unexpected finding, given our unique population with a high proportion of patients who had a history of injection drug use and substance use disorder [2]. Another study found significantly more DU-IE patients had valve surgery and tricuspid valve replacements compared with non-DU-IE patients [18]. Results from our study show that a higher proportion of patients who had surgery currently smoked. Smoking is thought to allow bacterial flora to enter the bloodstream through the oral tissue and can lead to the pathogenesis of infections such as IE [19]. Cigarette smoking has also been shown to redirect certain *Staphylococcus aureus* strains to virulent phenotypes associated with persistent infection [20].

Compared with patients who underwent surgery, a higher proportion of patients who were not surgically treated had comorbidities such as coronary artery disease, hyperlipidemia, acute kidney injury, and peripheral vascular disease. These results are similar to previous studies that showed higher comorbidities among patients who were not surgically treated [21, 22]. Surgery may be contraindicated in these patients due to either the severity or the number of comorbidities [22].

Our study found that compared to patients who did not have surgical treatment, significantly more patients with surgical treatment had mitral valve IE and aortic valve IE, but not tricuspid valve IE. This outcome may be due to worse hemodynamic tolerance in patients with mitral regurgitation-induced heart failure relative to tricuspid regurgitation-induced heart failure [22]. Findings from our study also show that several indications for surgery were significantly different between both groups, with more individuals who had surgery having vegetation size larger than 10 mm, pulmonary embolism, persistent sepsis/persistent positive blood cultures, or acute congestive heart failure, as is corroborated by studies [23].

Results from our study showed a significantly higher hospital utilization among surgically treated patients, both in terms of overall length of stay in the hospital and in the intensive care unit. Findings from other studies also corroborate the higher hospital utilization by patients who underwent surgical treatment [12]. However, compared to patients with surgery, more patients without surgery died during hospitalization. This finding is consistent with results from previous studies that reported higher in-hospital mortality in patients without surgical treatment [22, 24, 25]. A previous study observed the impact of early surgery in left-sided IE

and found that the overall mortality was 25% at 150 weeks, and that mortality was higher in non-surgically treated patients and those that refused surgery [25]. A large national study of approximately 35,000 valve surgeries for IE found that DU-IE is associated with higher in-hospital mortality (OR 1.15, 95% CI 1.01–1.31) [26].

We found a higher rate of discharge against medical advice among patients who did not undergo surgery. Previous studies have found that patients leave the hospital against medical advice due to negative staff interactions, poor management of withdrawal or pain, boredom, or isolation [27]. However, not completing the treatment increases the risk of life-threatening infections and poorer health outcomes overall [28, 29].

In this study, we found a significantly higher proportion of patients with surgery had injected drugs prior to hospital admission, including opioids, amphetamines, cannabinoids, cocaine metabolites, and benzodiazepines. DU-IE has been associated with injection of drugs such as opioids, methamphetamines, and cocaine [14]. Patients with DU-IE frequently require longer hospital stays and have a higher chance of potentially needing a reoperation [26]. A meta-analysis demonstrated significantly worse outcomes after cardiac surgery among IE patients who inject drugs compared to patients who do not inject drugs [16]. Even the 10-year outcomes, including mortality, recurrence, and reoperation were significantly higher among DU-IE patients who underwent surgery as shown in a prospective cohort study [30].

## Public health policy

The rising DU-IE in WV has unfolded new challenges of managing, treating, and caring for the patients. Due to the complexity in diagnosis and management of IE, especially DU-IE, interprofessional collaboration is crucial in the management of patients with IE. A multidisciplinary team including specialists in infectious diseases, cardiology, cardiac surgery, microbiology, and psychiatry can help with diagnosis, management, and treatment of patients with IE [31, 32]. Previous studies have shown that a multidisciplinary team can help improve overall outcomes in patients [33, 34].

Both the treatment complexity and the numerous relevant social determinants of health that influence risk of IE render urban and rural distinctions important with respect to both outcomes and risk factors. While attention must be placed on IE in both urban and rural settings, a deeper understanding of the treatment of rural IE is key as IE is increasing at a faster rate among rural patients [35]. The high burden of DU-IE and the right-sided IE in our study are likely due to regional factors, largely driven by high prevalence of drug use. For instance, counties without syringe services programs in Kentucky saw greater increases among DU-IE than those which had syringe services programs [36]. The treatment of IE for rural patients may also differ from that for urban patients because of greater distance to care, decreased access to home health or psychiatric resources, stigma regarding drug use, or other factors [37].

## Strengths and limitations

This study has limitations. Data on demographics such as education, income, and duration of drug use were not available. While studies have shown higher mortality rates post discharge, we were only able to show mortality rates among patients with IE while they were in the hospital. A longer-term follow-up of these patients would have further strengthened the study. In addition, since we were not able to collect the timelines for the risk factors, we are restricting our interpretation to associations only. Information on drug, alcohol, and cigarette use was mostly self-reported, and likely subject to self-reporting bias. However, one noteworthy advantage of this study is that it did not rely exclusively on ICD codes to retrieve patient information,

but on a complete manual chart review conducted for all patients. Another strength of this study includes its examination of individual patients rather than discharge databases.

## Conclusion

Surgery for IE has steeply increased in recent years, with the number of patients who underwent surgery increasing 26-fold between 2014 and 2018. A significantly higher number of patients with IE who had surgery were younger, currently smoked, had injected drugs and were on MOUD prior to hospital admission, were diagnosed with psychiatric disorders, and had more readmissions to hospitals as compared to those who did not have surgery. The decision to perform cardiac surgery on IE patients is complex, with outcomes varying by individual characteristics and several factors, including indications for surgery, resistance to antibiotics, prognosis of surgery, and the type of valve involved in IE. The current epidemic of injection drug use has exacerbated the need for a multidisciplinary team for comprehensive care of patients with IE.

## Supporting information

**S1 Checklist. STROBE statement—checklist of items that should be included in reports of observational studies.**
(DOCX)

## Author Contributions

**Conceptualization:** Ruchi Bhandari, Frank H. Annie, Umar Kaleem, Affan Irfan.

**Data curation:** Ruchi Bhandari, Frank H. Annie, Umar Kaleem, Ellen Thompson, Affan Irfan.

**Formal analysis:** Ruchi Bhandari, Noor Abdulhay, Talia Alexander.

**Funding acquisition:** Ruchi Bhandari.

**Investigation:** Ruchi Bhandari.

**Methodology:** Ruchi Bhandari.

**Project administration:** Ruchi Bhandari.

**Resources:** Ruchi Bhandari.

**Software:** Ruchi Bhandari.

**Supervision:** Ruchi Bhandari.

**Validation:** Ruchi Bhandari.

**Visualization:** Ruchi Bhandari.

**Writing – original draft:** Ruchi Bhandari, Noor Abdulhay, Jessica Rubenstein, Andrew Meyer, R. Constance Wiener, Cara Sedney.

**Writing – review & editing:** Ruchi Bhandari, Noor Abdulhay, Talia Alexander, Jessica Rubenstein, Andrew Meyer, Frank H. Annie, Umar Kaleem, R. Constance Wiener, Cara Sedney, Ellen Thompson, Affan Irfan.

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
