## [Decision Letter · Decision Letter 0]

4 Jun 2023

PONE-D-23-08921Characterization of patients receiving surgical versus non-surgical treatment for infective endocarditis in West VirginiaPLOS ONE

Dear Dr. Ruchi Bhandari,

Thank you for submitting your manuscript to PLOS ONE. After careful consideration, we feel that it has merit but does not fully meet PLOS ONE’s publication criteria as it currently stands. Therefore, we invite you to submit a revised version of the manuscript that addresses the points raised during the review process.

ACADEMIC EDITOR:

Dear authors, I would like to ask you address the comments given by the reviewers. 

We look forward to receiving your revised manuscript.

Kind regards,

Atnafu Mekonnen Tekleab, M.D

Academic Editor

PLOS ONE

Journal Requirements:

Reviewers' comments:

Reviewer's Responses to Questions

**Comments to the Author**

1. Is the manuscript technically sound, and do the data support the conclusions?

Reviewer #1: Partly

Reviewer #2: No

Reviewer #3: Yes

2. Has the statistical analysis been performed appropriately and rigorously? 

Reviewer #1: No

Reviewer #2: No

Reviewer #3: Yes

3. Have the authors made all data underlying the findings in their manuscript fully available?

Reviewer #1: Yes

Reviewer #2: No

Reviewer #3: No

4. Is the manuscript presented in an intelligible fashion and written in standard English?

Reviewer #1: Yes

Reviewer #2: Yes

Reviewer #3: Yes

5. Review Comments to the Author

Reviewer #1: This manuscript presents a retrospective review of endocarditis care and outcomes in at the 4 major cardiovascular medical centers in West Virginia between 2014 and 2018. The study is a result of significant effort which included hundreds of chart reviews to gather detailed information about characteristics of patients who do and do not receive surgery for endocarditis in West Virginia. There is a huge amount of information presented but the manuscript would be strengthened with improved framing. The introduction presents fairly general information about rising rates of endocarditis but does not set up the specific questions which extends into the presentation of the results and discussion. Specifically, introduction and discussion misses some key sign-posts that will help guide readers through the data. The data is stratified by receipt of surgery, which is also the outcome, but introduction doesn’t frame the manuscript around variation in surgical decision making or tensions and challenges. The high rate of DU-IE and relatively high rate of surgery in this cohort is unique and worth exploring further. Additionally, the model selection process requires greater explanation. Finally, in-hospital opioid use disorder treatment strategy and/or at time of discharge is conspicuously absent given the frame of the paper and importance to this issue.

Major:

1. Abstract: The abstract could better set up the question addressed about predicotrs of surgery. Suggest using specific terminology rather than “outcomes.” This study looks at predictors of surgery and reports on inpatient mortality and readmission.

2. Abstract: Unhealthy opioid use is vague terminology which may or may not influence risk for endocarditis. Suggest using more specific terminology – injection drug use.

3. Abstract conclusion: The conclusion is quite general and should more clearly highlight the key findings.

4. When discussing changing approaches to managing IDU, consider citing this recent scientific statemenet which highlights some of the key tensions and challenges in caring for this group of patients: Baddour et al. Management of infective endocarditis in people who inject drugs: a scientific statement from the American Heart Association. Circulation. 2022;146:e187–e201.

5. Drug use is a key covariate but the broad category seems to elide conditions which significantly increase risk of endocarditis from injection and those that do not (such as cannabis).

6. For patients categorized as having drug use (positive for buprenorphine or methadone), I presume since there is a separate category MOUD, does this mean non-prescribed use? How about benzos?

7. For the outcome surgery, is this definied exclusively as surgery during the index admission? Would someone who is admitted, leaves ama, and is readmitted and then receives surgery be included in this cohort?

8. The term substance use disorder is used in table 2 and drug use in table 1. In this data, hwo are these defined and different.

9. Results: In writing the comparison with parenthesis, the order of surgery vs non-surgery goes back and forth. For clarity, this should be consistnet.

10. Can the authors report the number of hospital days until surgery was performed?

11. For outcomes, is readmission for any reason or endocarditis related? Additionally, is the follow up time consistent across individuals or variable. For example, someone who enters the cohort at the beginning, has 4 years of potential follow up time where as someone later in the cohort could have very little follow up time.

12. Multivariable modeling. I do not fully understand the selection of variables to include in the model. There are theoretically informed and quantitative approaches, each with it’s own strengths and weaknesses. The authors seem to state that variables which were statistically significant in the bivariate models were included. However, AV and MV endocarditis are more likely to be in the surgery group but are not included. Further, there are difference by year that is not included. Based on both theoretical and quantitative approaches, it seems very important to control for left vs right-side disease and year in this data.

13. The manuscript seems to tell a story of rising endocarditis, driven by drug use, but also rising surgical rates (row percents in table 1 by year would show that clearly). I am left wondering why surgical rates are increasing in this cohort. Is there more left sided disease over time? Is surgical decision making changing – ie. have distribution of surgical indications changed? The authors should consider exploring these trends. Clinically speaking, I think these question are central and under-explored

14. Discharge status categories are confusing. AMA discharge patients are discharged alive but the categories are mutually exclusive. Is location of discharge available as well (home with services, to a facility, etc)

15. The central narrative of this manuscript is that drug use is driving endocarditis in West Virginia but how drug use is addressed in the cohort is not reported. Are patients offered MOUD in the hospital with linkage to care? If not, how is withdrawal managed? What addiction treatment resources are offered or referred?

16. Discussion: Line 301/302, the authors state that people who use drugs have higher in-hospital mortality rates but the citation reports only on 2 yr mortality. The authors should be careful when citing and generalizing. In fact, several studies actually show that among all patients with endocarditis, in-hosptial mortality is lower for people with DU-IE compared to non-DU-IE.

Minor:

1. Consider editing short title to, “Characteristics of endocarditis patients in West Virginia” as the study reports on those who do and do not have surgery.

2. Introduction: page 4, line 94. The statement on repair vs replacement “minimizing the penetration of for materials” is awkwardly worded. Further, the citation does not support such a strong statement. Consider softening the language.

3. In Table 2 under valve, the aortic p-value is bolded (presumably to represent statistical significance) but mitral valve is not.

Reviewer #2: This paper on Characterization of patients receiving surgical versus non-surgical treatment for infective endocarditis in West Virginia addresses an important issue with a rising prevalence in the area. It describes the surgical characteristics and outcomes of patients with IE who received medical treatment alone or those who received both medical and surgical treatment in rural centers in West Virginia and makes comparison among the groups.

This article could be improved if the methods and result sections are revised and the discussion section is also addressed after that. The methodological issues that may lead to biases also need to be discussed as there are a number of baseline differences between the two comparison groups and from the design of the study.

The methodology section is not sufficient.

There authors did not mention whether the study conforms to any relevant guidelines.

The outcome variables not clearly defined. The objective of the paper as outlined at the end of introduction section is “to characterize the outcomes among patients hospitalized with IE stratified by those who received only antimicrobial treatment versus those who received antimicrobial and surgical treatment in the four major rural centers in West Virginia.” How was that addressed in the analysis?

Operational definitions are missing

How was the diagnosis of infective endocarditis made?

How was sample size calculated? What is the power of the study?

Result

Line 169-171 says “During this period, there was a 2.4-fold increase in patients who were treated non-surgically and an 18-fold increase in patients who underwent surgery as part of their management (p < 0.001)”. How or between which years was the comparison done?

How did you take the reference when calculation for association (significance) was done for

Table 1? The way the associations were described in the table lacks uniformity.

“Total” is missing and some of the numbers (percents) do not add up to 100. It would be difficult to make any interpretation from the tables as they are now. This needs revision.

Significant difference in the baseline characteristics?

The same is true for table 2

Figure 1: blurred image

Line 191-194: A significantly higher proportion of patients without surgery had comorbidities, including …. while a high proportion of patients with surgery were diagnosed with psychiatric disorders.

Table 3: There are quite a number of patients who were discharged within short days of hospital stay. How is this explained in terms of the duration of hospitalization or antibiotic treatment for IE?

What is the relevance of Table 4?

Table 5 needs more description. What is explained is the presence of associations only.

Discussion and conclusion need to be revised after addressing the methods and result section and has to try to answer the objectives of the study. The methodological biases or flaws that may arise from the nature of the study design or from the characteristics of the participants in the study also need to be addressed in the limitation section.

Reviewer #3: Dear Ruch Bhandari et al.,

Thanks for bringing a paper of public health importance, infective endocarditis.

Your paper highlights key issues among IE adult subjects who were treated medically and medically-surgically.

Please find here comments, suggestions and questions forwarded to help improve your manuscript.

Abstract:

Background: page 2 line 47 ‘the purpose….is to characterize’ better if worded in past tense ‘the purpose…was to characterize’

Results: Please provide p values for univariate analysis outputs which showed difference between categories e.g., age, drug use, psychiatric disorders…etc. Same is true for death and discharge against medical advice.

Conclusion: it would be good if authors could suggest some clinical recommendations based on the study findings.

Background

Repetition of idea: ‘West Virginia had a 681% increase in overall IE hospitalizations in the

104 state between 2014 and 2018, predominantly associated with injection drug use’. Check if one of these can be omitted or if it a must for you to keep both try rewording it.

Methods

Please include brief background information on the cardiovascular rural centers in the study setting.

Please include operational definition for persistent sepsis/ persistent positive blood culture. Please include inclusion and exclusion criteria for this retrospective review.

Please clarify or mention which variables were included in the final model and how was selection made for this.

Few comments on the sample size and power of your study to bring recommendations.

Results

Page 11: ‘what was the reason for inclusion of the multispecialty consultation services?’ on line 08-211. If it is peculiar for the study subjects then further elaborations need to be provided.

Discussion

Summarized well. Please further expand and include clinical recommendations on IE management based on your study.

References: appropriate

Abbreviations: please check some abbreviations as some are not first written in full like AMA ….

Best regards,

6. PLOS authors have the option to publish the peer review history of their article (what does this mean?). If published, this will include your full peer review and any attached files.

Reviewer #1: No

Reviewer #2: No

Reviewer #3: **Yes: **Henok Tadele

---

## [Author Response · Author response to Decision Letter 0]

21 Jul 2023

Major:

1. Reviewer Comment: Abstract: The abstract could better set up the question addressed about predicotrs of surgery. Suggest using specific terminology rather than “outcomes.” This study looks at predictors of surgery and reports on inpatient mortality and readmission. 

Author Response: Thank you for your suggestion. We have revised the abstract per your suggestions in Lines: 42 – 83. 

2. Reviewer Comment: Abstract: Unhealthy opioid use is vague terminology which may or may not influence risk for endocarditis. Suggest using more specific terminology – injection drug use. 

Author Response: Thank you. We have modified the terminology to injection drug use in the abstract.

3. Reviewer Comment: Abstract conclusion: The conclusion is quite general and should more clearly highlight the key findings. 

Author Response: Thank you for your suggestion. We have modified the conclusion with the given constraint of 300-word limit. 

4. Reviewer Comment: When discussing changing approaches to managing IDU, consider citing this recent scientific statemenet which highlights some of the key tensions and challenges in caring for this group of patients: Baddour et al. Management of infective endocarditis in people who inject drugs: a scientific statement from the American Heart Association. Circulation. 2022;146:e187–e201. 

Author Response: We have now added the suggested reference in Line: 124.

5. Reviewer Comment: Drug use is a key covariate but the broad category seems to conditions which significantly increase risk of endocarditis from injection and those that do not (such as cannabis). 

Author Response: Yes, we have now revised the language to include injection drug use all through the manuscript. Of the patients who reported drug use, 97% reported injecting drugs. However, given the history of drug use and the current diagnosis of infective endocarditis, and after consultation with infectious disease specialist, we have included the remaining 3% also in this category since they had a history of using drugs and were diagnosed with IE. 

6. Reviewer Comment: For patients categorized as having drug use (positive for buprenorphine or methadone), I presume since there is a separate category MOUD, does this mean non-prescribed use? How about benzos? 

Author Response: Yes, it means non-prescribed use. 

7. Reviewer Comment: For the outcome surgery, is this definied exclusively as surgery during the index admission? Would someone who is admitted, leaves ama, and is readmitted and then receives surgery be included in this cohort? 

Author Response: Yes, surgery is defined as surgery during the index admission. Only IE patients with index surgery are included in this study. So, no, someone who is admitted, leaves AMA, and is readmitted and then receives surgery will NOT be included in this cohort. We have clarified in Lines 157 – 158: The outcome variable is patients with IE undergoing surgery (along with antimicrobial treatment) vs. no surgery (antimicrobial treatment alone) at patient’s index admission for IE.

8. Reviewer Comment: The term substance use disorder is used in table 2 and drug use in table 1. In this data, hwo are these defined and different.

Author Response: Not all patients with drug use have been diagnosed with substance use disorder. Our previous paper using the same patient population shows that 40.69% of the IE patients who used drugs were diagnosed with SUD. (Bhandari R, Alexander T, Annie FH, Kaleem U, Irfan A, Balla S, et al. Steep rise in drug use-associated infective endocarditis in West Virginia: characteristics and healthcare utilization. PLOS One. 2022 Jul;17(7):e0271510. https://doi.org/10.1371/journal.pone.0271510 PMID:35839224).

9. Reviewer Comment: Results: In writing the comparison with parenthesis, the order of surgery vs non-surgery goes back and forth. For clarity, this should be consistnet. 

Author Response: Thank you for the helpful suggestion. All through the manuscript, we now have “surgery” first and “no surgery” later in the sentence. For the same reason, we have now modified our tables to include “surgery” first and “no surgery” later.

10. Reviewer Comment: Can the authors report the number of hospital days until surgery was performed? 

Author Response: Sorry, we did not collect that data. It would have been very useful.

11. Reviewer Comment: For outcomes, is readmission for any reason or endocarditis related? Additionally, is the follow up time consistent across individuals or variable. For example, someone who enters the cohort at the beginning, has 4 years of potential follow up time where as someone later in the cohort could have very little follow up time. 

Author Response: Readmission is for endocarditis only. The database contains patients who had their index admission for IE during January 2014 and December 2018. The only follow-up is the first readmission (yes/no) anytime during the study period after the patient’s discharge. 

12. Reviewer Comment: Multivariable modeling. I do not fully understand the selection of variables to include in the model. There are theoretically informed and quantitative approaches, each with it’s own strengths and weaknesses. The authors seem to state that variables which were statistically significant in the bivariate models were included. However, AV and MV endocarditis are more likely to be in the surgery group but are not included. Further, there are difference by year that is not included. Based on both theoretical and quantitative approaches, it seems very important to control for left vs right-side disease and year in this data. 

Author Response: Thank you for your important advice. We have now revised the multivariable analysis and incorporated the two variables (AV/MV IE and Year) in the model as you suggested (Lines 186-188). However, given the smaller sample size and lack of bivariate association, with the addition of these two variables, we have excluded comorbidities and organism type. The revised Table 5 appears on Page 19.

13. Reviewer Comment: The manuscript seems to tell a story of rising endocarditis, driven by drug use, but also rising surgical rates (row percents in table 1 by year would show that clearly). I am left wondering why surgical rates are increasing in this cohort. Is there more left sided disease over time? 

Author Response: Yes, Figure 1 shows the rising surgical rates per year. The rising surgical rates are a result of rising injection drug use-associated endocarditis (reference: (Bhandari R, Alexander T, Annie FH, Kaleem U, Irfan A, Balla S, et al. Steep rise in drug use-associated infective endocarditis in West Virginia: characteristics and healthcare utilization. PLOS One. 2022 Jul;17(7):e0271510. https://doi.org/10.1371/journal.pone.0271510 PMID:35839224), early surgical interventions, and access to appropriate healthcare. 

Regarding the changes in valves affected over the years, I ran the analysis and found the following results for percentage of patients with each affected valve in 2014 and 2018: 

Affected Valve 2014 2018

Aortic valve 19.1% 21.6%

Mitral valve 38.3% 24.8%

Tricuspid valve 38.3% 50.7%

Pulmonic valve 4.3% 4.0%

So, your conjecture is correct. Composition of patients with Mitral valve IE has decreased and Tricuspid valve IE has increased significantly.

Reviewer Comment: Is surgical decision making changing – ie. have distribution of surgical indications changed? The authors should consider exploring these trends. Clinically speaking, I think these question are central and under-explored.

Author Response: Yes, the percentage of patients with higher number of indications for surgery has increased between 2014 and 2018. The increase in the following indications for surgery was statistically significant: Persistent sepsis/persistent positive blood cultures, Embolism, and Stroke.

We have reviewed literature from past few years and did not notice major difference in surgical decision making. One common theme that is more pronounced now is the need for multidisciplinary comprehensive care by the endocarditis team. There is significant clinical judgment for the individual patient and the hospital. As in the following reference, there are “areas of uncertainty and gaps in current evidence for the use of surgery in IE across different indications.” (Wang A, Fosbøl EL. Current recommendations and uncertainties for surgical treatment of infective endocarditis: a comparison of American and European cardiovascular guidelines. European heart journal. 2022:43(17), 1617–1625. https://doi.org/10.1093/eurheartj/ehab898)

14. Reviewer Comment: Discharge status categories are confusing. AMA discharge patients are discharged alive but the categories are mutually exclusive. Is location of discharge available as well (home with services, to a facility, etc). 

Author Response: AMA and discharge alive are mutually exclusive categories. Only those who were not AMA and did not die, were discharged alive. We have added this sentence in the Methods section Line 168: The discharge categories are mutually exclusive.

Following information is available in a previous paper (Reference: (Bhandari R, Alexander T, Annie FH, Kaleem U, Irfan A, Balla S, et al. Steep rise in drug use-associated infective endocarditis in West Virginia: characteristics and healthcare utilization. PLOS One. 2022 Jul;17(7):e0271510. https://doi.org/10.1371/journal.pone.0271510 PMID:35839224). 

Discharge location N % 

Home 521 66.71

Skilled/other nursing facility 56 7.17

Another hospital/facility 52 6.66

Residential substance use treatment facility 33 4.35

Acute rehab 17 2.18

Long-term acute care facility 16 2.05

Missing 10 1.28

15. Reviewer Comment: The central narrative of this manuscript is that drug use is driving endocarditis in West Virginia but how drug use is addressed in the cohort is not reported. Are patients offered MOUD in the hospital with linkage to care? If not, how is withdrawal managed? What addiction treatment resources are offered or referred? 

Author Response: Yes, the hospitals in this study offer MOUD to patients with opioid use disorder (OUD). As soon as the patient is medically stable, hospitals attempt to deal with withdrawal through various non-pharmacological and pharmacological approaches, such as supportive care, buprenorphine, or methadone. Attempts are made to establish a linkage to outpatient care, e.g., short- or long-term rehab facility, to ensure continuity of outpatient addiction care and support for the patient after discharge.

However, our previous paper “found that 40% of patients who had received MOUD before admission were not prescribed MOUD during hospitalization. This represents a missed opportunity to provide continued SUD care that may reduce the risk of recidivism.” (Reference: (Bhandari R, Alexander T, Annie FH, Kaleem U, Irfan A, Balla S, et al. Steep rise in drug use-associated infective endocarditis in West Virginia: characteristics and healthcare utilization. PLOS One. 2022 Jul;17(7):e0271510. https://doi.org/10.1371/journal.pone.0271510 PMID:35839224). 

16. Reviewer Comment: Discussion: Line 301/302, the authors state that people who use drugs have higher in-hospital mortality rates but the citation reports only on 2 yr mortality. The authors should be careful when citing and generalizing. In fact, several studies actually show that among all patients with endocarditis, in-hosptial mortality is lower for people with DU-IE compared to non-DU-IE. 

Author Response: We apologize for the generalization with the above reference. We agree that a few studies have shown lower DU-IE in-hospital mortality. We have revised the sentence in Line 344-346 and the reference: A large national study of approximately 35,000 valve surgeries for IE found that DU-IE is associated with higher in-hospital mortality (OR 1.15, 95% CI 1.01–1.31). Reference 26 is now: Geirsson A, Schranz A, Jawitz O, Mori M, Feng L, Zwischenberger BA, Iribarne A, Dearani J, Rushing G, Badhwar V, Crestanello JA. The Evolving Burden of Drug Use Associated Infective Endocarditis in the United States. Ann Thorac Surg. 2020 Oct;110(4):1185-1192. doi: 10.1016/j.athoracsur.2020.03.089. PMID: 32387035. 

Minor:

1. Reviewer Comment: Consider editing short title to, “Characteristics of endocarditis patients in West Virginia” as the study reports on those who do and do not have surgery.

Author Response: Thank you. We have revised the short title: Characteristics of patients with and without endocarditis surgery.

2. Reviewer Comment: Introduction: page 4, line 94. The statement on repair vs replacement “minimizing the penetration of for materials” is awkwardly worded. Further, the citation does not support such a strong statement. Consider softening the language. 

Author Response: We have now revised the sentence in Lines 108-109: In patients who inject drugs, valve repair is preferred over replacement to preserve valve function and avoid foreign material implant. We have also changed the reference to Baddour et al, 2022 (Reference number 3).

3. Reviewer Comment: In Table 2 under valve, the aortic p-value is bolded (presumably to represent statistical significance) but mitral valve is not. 

Author Response: Thank you. We have added this Note under Tables 1-3 on Pages 8 - 16: Highlighted P-values reflect statistical significance after Bonferroni correction.

Reviewer #2: 

Reviewer Comment: This paper on Characterization of patients receiving surgical versus non-surgical treatment for infective endocarditis in West Virginia addresses an important issue with a rising prevalence in the area. It describes the surgical characteristics and outcomes of patients with IE who received medical treatment alone or those who received both medical and surgical treatment in rural centers in West Virginia and makes comparison among the groups.

Author Response: Thank you, West Virginia is truly experiencing the dual epidemic of injection drug use and associated severe infectious diseases.

Reviewer Comment: This article could be improved if the methods and result sections are revised and the discussion section is also addressed after that. The methodological issues that may lead to biases also need to be discussed as there are a number of baseline differences between the two comparison groups and from the design of the study.

The methodology section is not sufficient.

Author Response: Thank you. We have now revised the methods and results sections according to specific comments by all reviewers. 

This study is not a sample, but all index patients admitted to the four hospitals for infective endocarditis during 2014-2018 were included in the study.

Yes, we agree there are differences in the patients with and without surgery. We want to highlight those differences as shown in Tables 1-4 (Pages 8 - 18) that are descriptive analyses of all the patients by their surgery status. To minimize bias, we included the multivariable regression in Table 5 (Page 18-19) and controlled for the important potential confounders. However, given the sample size, we needed to limit the number of potential confounders in the model.

In the Discussion section in Lines 396 - 397, we have the following sentence as a limitation of our study: Information on drug, alcohol, and cigarette use was mostly self-reported, and likely subject to self-reporting bias.

In addition, we have included the following sentence in Lines 394 - 396: In addition, since we were not able to collect the timelines for the risk factors, we are restricting our interpretation to associations only.

Reviewer Comment: There authors did not mention whether the study conforms to any relevant guidelines. 

Author Response: Thank you. Yes, we agree the “STrengthening the Reporting of Observational studies in Epidemiology” (STROBE) guidelines are very important for observational studies. Therefore, we had submitted the STROBE checklist for our manuscript and the editor may be willing to share with you if you would like to see.

Reviewer Comment: The outcome variables not clearly defined. The objective of the paper as outlined at the end of introduction section is “to characterize the outcomes among patients hospitalized with IE stratified by those who received only antimicrobial treatment versus those who received antimicrobial and surgical treatment in the four major rural centers in West Virginia.” How was that addressed in the analysis?

Author Response: To characterize the outcomes among patients hospitalized with IE stratified by those who received only antimicrobial treatment versus those who received antimicrobial and surgical treatment in the four major rural centers in West Virginia, we have presented descriptive statistics by surgery status, and results from multivariable logistic regression analysis that examined the association of surgery with key predictor variables. Tables 1-5 (Pages 8 - 19).

Reviewer Comment: Operational definitions are missing 

Author Response: We apologize for the oversight. We have now included the sentence in Lines 157 – 158: The outcome variable is patients with IE undergoing surgery (along with antimicrobial treatment) vs. no surgery (antimicrobial treatment alone) at patient’s index admission for IE.

Reviewer Comment: How was the diagnosis of infective endocarditis made? 

Author Response: Thank you. Lines 143-145 have the following information: Patients were identified initially using the International Classification of Diseases, Tenth Revision, Clinical Modification (ICD-10-CM) codes for IE, followed by a manual chart review for all admissions to ensure accurate diagnosis confirmation. 

Reviewer Comment: How was sample size calculated? What is the power of the study? 

Author Response: Thank you for your question. This is not a sample. It is all the patients in the multi-site study admitted for infective endocarditis.

Taking drug use as the variable influencing the odds of undergoing surgery controlling for other confounders, we conducted the sample size calculation. For this analysis, a sample size calculation for a logistic regression, assuming a two-sided test, an effect size (odds ratio) of 1.5, a probability of surgery under the null hypothesis of 0.20, alpha of 0.05, power of 80%, and moderate variance explained among other covariates (R2=0.25, revealed that 410 total participants would be required. There were 779 patients included in this analysis, which far exceeds the sample size required. Furthermore, we assumed a lower odds ratio than was observed and a lower probability of surgery than was observed. 

Reviewer Comment: Result Line 169-171 says “During this period, there was a 2.4-fold increase in patients who were treated non-surgically and an 18-fold increase in patients who underwent surgery as part of their management (p < 0.001)”. How or between which years was the comparison done? 

Author Response: Thank you for your question. Lines 194-197 state: “Of the 780 patients with IE who were admitted between January 1, 2014, and December 31, 2018, 37.9% had surgery. During this period, there was a 2.4-fold increase in patients who were treated non-surgically and an 18-fold increase in patients who underwent surgery as part of their management (p < 0.001) (Fig 1).”

Reviewer Comment: How did you take the reference when calculation for association (significance) was done for Table 1? The way the associations were described in the table lacks uniformity. 

Author Response: As stated in the Methods section: Lines 176 - 178: Surgery and non-surgery groups were compared using Chi-square test or Fisher’s exact test when expected cell count was <5. Continuous variables are presented as median and interquartile range. 

Table 1 compares patient characteristics between the surgery vs no surgery groups, which is formatted in the journal PLOS ONE's style requirements. 

Reviewer Comment: “Total” is missing and some of the numbers (percents) do not add up to 100. It would be difficult to make any interpretation from the tables as they are now. This needs revision. Significant difference in the baseline characteristics? The same is true for table 2

Author Response: Thank you. We have revised our Tables 1 – 3 (Pages 8 - 16) so the values equal 100%. Variables with individual p-values (e.g., different types of drugs in Table 1, and all variables in Table 2), are separate variables. Chi-square test was done to compare each variable between surgery vs no surgery. We have also added a footnote to the Tables that p-values reflect statistical significance after Bonferroni correction. 

Reviewer Comment: Figure 1: blurred image 

Author Response: We have resubmitted a clear Figure, following the journal’s guidelines. We would request the Editor to please share the clearer image with the reviewer.

Reviewer Comment: Line 191-194: A significantly higher proportion of patients without surgery had comorbidities, including …. while a high proportion of patients with surgery were diagnosed with psychiatric disorders.

Author Response: Thank you for your comment. All through the text, we have now revised our results so that surgery results appear first. Specifically, lines 219 – 226 are now revised thus: “A significantly lower proportion of patients with surgery had comorbidities, including Type 2 diabetes, coronary artery disease, hyperlipidemia, acute kidney injury, and peripheral vascular disease compared to patients without surgery. However, a higher proportion of patients with surgery were diagnosed with psychiatric disorders (57.3% vs. 31.0%) including, substance use disorder (45.1% vs. 19.0), depression (28.8% vs 15.9%), and anxiety (27.5% vs. 12.4%) (all p < 0.001), compared to patients without surgery.”

Reviewer Comment: Table 3: There are quite a number of patients who were discharged within short days of hospital stay. How is this explained in terms of the duration of hospitalization or antibiotic treatment for IE? 

Author Response: The length of stay is the actual length of hospitalization obtained from the medical records. The patients are given antibiotic treatment and surgery, if needed, during their hospitalization. 

Reviewer Comment: What is the relevance of Table 4? 

Author Response: Table 4, Page 17 – 18 presents the surgical characteristics of patients with infective endocarditis, such as 80% of the patients who underwent surgery had cardioplegia, or 40% had sinus rhythm. 

Reviewer Comment: Table 5 needs more description. What is explained is the presence of associations only.

Author Response: Yes, we agree with you that since we do not have the timelines for the risk factors, we should interpret the results as associations only. As stated in lines 273-278: In the multivariable logistic regression results presented in Table 5, surgery was statistically significantly associated with injection drug use (OR: 1.895; 95% CI: 1.089-3.299), left-sided valve with reference to right-sided valve (2.136; 95% CI: 1.429-3.193), bilateral valve with reference to right-sided valve (2.473; 95% CI: 1.240-4.932), number of indications for surgery (OR: 1.680; 95% CI: 1.482-1.905), year 2017 with reference to 2014-2016 (OR: 3.147; 95% CI: 2.030-4.877) and year 2018 with reference to 2014-2016 (OR: 3.746; 95% CI: 2.455-5.716).

Reviewer Comment: Discussion and conclusion need to be revised after addressing the methods and result section and has to try to answer the objectives of the study. The methodological biases or flaws that may arise from the nature of the study design or from the characteristics of the participants in the study also need to be addressed in the limitation section. 

Author Response: Thank you. We would like to keep the focus of this paper as epidemiologic characterization of IE patients in West Virginia within the field of public health epidemiology per the objectives of the study. To that extent, we have presented and discussed our results at length. Given the dual epidemic of injection drug use and associated infectious diseases, we strongly recommend that the clinical guidelines include the need for multidisciplinary comprehensive team approach to assess and care for the infection as well as the addiction. Please see the “Public Health Policy’” section Lines 369 – 387.

We also have the following lines on limitations of the study (Lines 391 - 397): This study has limitations. Data on demographics such as education, income, and duration of drug use were not available. While studies have shown higher mortality rates post discharge, we were only able to show mortality rates among patients with IE while they were in the hospital. A longer-term follow-up of these patients would have further strengthened the study. In addition, since we were not able to collect the timelines for the risk factors, we are restricting our interpretation to associations only. Information on drug, alcohol, and cigarette use was mostly self-reported, and likely subject to self-reporting bias. 

Reviewer #3: Dear Ruch Bhandari et al.,

Thanks for bringing a paper of public health importance, infective endocarditis.Your paper highlights key issues among IE adult subjects who were treated medically and medically-surgically. Please find here comments, suggestions and questions forwarded to help improve your manuscript.

Reviewer Comment: 

Abstract:

Background: page 2 line 47 ‘the purpose….is to characterize’ better if worded in past tense ‘the purpose…was to characterize’ 

Results: Please provide p values for univariate analysis outputs which showed difference between categories e.g., age, drug use, psychiatric disorders…etc. Same is true for death and discharge against medical advice. 

Conclusion: it would be good if authors could suggest some clinical recommendations based on the study findings. 

Author Response: Thank you. We have revised the abstract per all your helpful comments. 

Reviewer Comment: Background Repetition of idea: ‘West Virginia had a 681% increase in overall IE hospitalizations in the 104 state between 2014 and 2018, predominantly associated with injection drug use’. Check if one of these can be omitted or if it a must for you to keep both try rewording it. 

Author Response: Thank you, we have deleted the second sentence in Line 119 – 121. 

Reviewer Comment: Methods Please include brief background information on the cardiovascular rural centers in the study setting. 

Author Response: We have now added the following sentence in Lines 141 - 143 describing the hospitals: The hospitals in the study are the only tertiary centers in the state with the capability to perform heart surgery, and patients are referred to these centers from across the entire state.

Reviewer Comment: Please include operational definition for persistent sepsis/ persistent positive blood culture. 

Author Response: Per current guidelines, “ESC and ACC/AHA guidelines both recommend surgery in the setting of a highly resistant organism, evidence of abscess or penetrating lesion, or persistent bacteremia; all predictive of persistent infection with antibiotic therapy alone.) (Reference: https://www.acc.org/Latest-in-Cardiology/ten-points-to-remember/2022/05/09/19/39/Current-Recommendations-and-Uncertainties)

Our Infectious Disease specialist answered that the surgeons and ID doctors do not use a strict clinical definition for “persistent positive blood culture.” They generally use the term when a patient continues to have positive blood cultures for around five days or more while on optimal antimicrobial therapy. 

Reviewer Comment: Please include inclusion and exclusion criteria for this retrospective review. 

Author Response: Thank you. We have now added the inclusion criteria in Line 137 -145: This study is a retrospective chart review of EHR of all adults between the ages of 18 to 90 years who had their first IE hospital admission between January 1, 2014, and December 31, 2018, in four university-affiliated referral hospitals in West Virginia. The hospitals in the study are the only tertiary centers in the state with the capability to perform heart surgery, and patients are referred to these centers from across the entire state [2]. Patients were identified initially using the International Classification of Diseases, Tenth Revision, Clinical Modification (ICD-10-CM) codes for IE [2], followed by a manual chart review for all admissions to ensure accurate diagnosis confirmation.

Reviewer Comment: Please clarify or mention which variables were included in the final model and how was selection made for this. 

Author Response: Thank you. Based on another reviewer’s comments, we have revised the final multivariable model using the following variables and included in the methods section, lines 184 - 189: Multivariable logistic regression analysis was conducted to examine the association between the key dependent variable, surgery (yes/no), and key predictor and potentially confounding variables: age (continuous), injection drug use (dichotomous), valve (right-sided/left-sided/bilateral), number of indications for surgery (ordinal), and years (2014-2016/2017/2018). Other variables, such as sex and race, were not included in the analysis because they were not significant in the bivariate analysis.

Reviewer Comment: Few comments on the sample size and power of your study to bring recommendations.

Author Response: Thank you. As responded to an earlier reviewer, this is not a sample. It is all the patients in the multi-site study admitted for infective endocarditis.

However, per the reviewer request, we are sharing results from sample size calculation. Taking drug use as the variable influencing the odds of undergoing surgery controlling for other confounders, we conducted the sample size calculation. For this analysis, a sample size calculation for a logistic regression, assuming a two-sided test, an effect size (odds ratio) of 1.5, a probability of surgery under the null hypothesis of 0.20, alpha of 0.05, power of 80%, and moderate variance explained among other covariates (R2=0.25), revealed that 410 total participants would be required. There were 779 patients included in this analysis, which far exceeds the sample size required.

Reviewer Comment: Results: Page 11: ‘what was the reason for inclusion of the multispecialty consultation services?’ on line 08-211. If it is peculiar for the study subjects then further elaborations need to be provided. 

Author Response: Thank you. Following the characterization of patients with and without surgery, we provided this information in lines 240-243: Moreover, a higher proportion of patients with surgery also utilized the following consultation services: cardiac surgery, general surgery, interventional radiology, dentistry, pain management, social work, physical/occupational therapy, and psychiatry.

This finding indicates the need for comprehensive multidisciplinary care, possibly even more so among those who undergo surgery.

Reviewer Comment: Discussion: Summarized well. Please further expand and include clinical recommendations on IE management based on your study.

Author Response: Thank you. We would like to keep the focus of this paper as epidemiologic characterization of IE patients in West Virginia within the field of public health epidemiology. To that extent, we have presented and discussed our results at length. Given the dual epidemic of injection drug use and associated infectious diseases, we strongly recommend that the clinical guidelines include the need for multidisciplinary comprehensive team approach to assess and care for the infection as well as the addiction. Please see the “Public Health Policy’” section Lines 369 – 387.

Reviewer Comment: References: appropriate

Reviewer Comment: Abbreviations: please check some abbreviations as some are not first written in full like AMA 

Author Response: Thank you very much and we apologize for having missed a few. We have now ensured all abbreviations first appear in full form. 

Comment to the editor: 

We have included new Tables 1-3 and crossed out the old tables in Track Changes. This is because we moved the column “Surgery” before “No surgery” based on the comment from Reviewer 1. We also rounded off the numbers to one decimal point per the journal’s requirement. We have also revised Table 1 per the suggestion by Reviewer 1. We greatly appreciate all reviewers for their time and valuable feedback which, we think, has strengthened and improved the paper.

---

## [Editor Report · Decision Letter 1]

24 Jul 2023

Characterization of patients receiving surgical versus non-surgical treatment for infective endocarditis in West Virginia

PONE-D-23-08921R1

Dear Dr. Bhandari,

We’re pleased to inform you that your manuscript has been judged scientifically suitable for publication and will be formally accepted for publication once it meets all outstanding technical requirements.

Kind regards,

Atnafu Mekonnen Tekleab, M.D

Academic Editor

PLOS ONE
---

## [Editor Report · Acceptance letter]

7 Sep 2023

PONE-D-23-08921R1 

Characterization of patients receiving surgical versus non-surgical treatment for infective endocarditis in West Virginia 

Dear Dr. Bhandari:

I'm pleased to inform you that your manuscript has been deemed suitable for publication in PLOS ONE. Congratulations! Your manuscript is now with our production department. 

Kind regards, 

on behalf of

Dr. Atnafu Mekonnen Tekleab 

Academic Editor

PLOS ONE